# The Impacts of External Sustainability: Institutional Investors' Sustainable Identity, Corporate Environmental Responsibility, and Green Innovation

Xiao Yan [1] and Chengning Yang [2,*]

[1] School of Economics and Management, Inner Mongolia Normal University, Hohhot 010011, China; 20200020@imnu.edu.cn
[2] School of Finance and Business, Shanghai Normal University, Shanghai 200234, China
[*] Correspondence: yangchengning@shnu.edu.cn; Tel.: +86-139-1713-7742

**Abstract:** Motivated by the growing importance of corporate sustainable development and corporate executives' strong desire for shareholder input, this paper fulfills the research gap of corporate green innovation determinants from the view of institutional investors' sustainability, which is scarcely investigated in related research. Prior research (on green innovation determinants) mostly focused on internal sustainability's influencing effects (e.g., green absorptive capacity, green organizational identify); few investigated the role of external sustainability (e.g., institutional investors) in influencing corporate green innovation. We examine the potential impact of institutional investors' sustainable identity and corporate environmental responsibility efforts on green innovation, utilizing the difference-in-differences (DID) design along with Chinese-listed companies' data from 2010 to 2020. Our empirical results confirm that an institutional investor's sustainable identity has a promoting effect on corporate green innovation. This promoting effect is more pronounced in companies that perform better in environmental responsibility. Our cross-sectional analysis validates such better-performing effects. Additionally, we find that this external sustainable identity produces a shock effect similar to a sustainable rating from a third-party agency on corporate green innovation. Our study contributes to the literature on green innovations' external green (sustainable) determinants and the research on institutions' outcomes (prior research investigated institutional investors' various characteristics, such as ownership dispersion and site visit, on influencing corporate green innovation, though few determined whether their sustainable identity produced such effects).

**Keywords:** institutional sustainable identity; environmental responsibility performance; green innovation; staggered difference-in-difference method; China

## 1. Introduction

Global climate change and environmental pollution are issues of increasing concern to humanity. As a result, companies are working toward developing a more ethical and sustainable path. China, the largest manufacturing country worldwide, is facing serious energy and environmental issues. In the "2018 Environmental Performance Index" (Yale University), China ranked 177 among 180 countries in air quality indicators. Thus, it is advised that Chinese companies fulfill their environmental and social responsibility to concentrate on green behavior and sustainable growth.

Green innovation is one of the main strategies for companies to achieve sustainable development [1–5]. Numerous prior works analyzed its determinants from both internal and external perspectives, such as organizational green absorptive capacity [6], green organizational identity [7], board gender diversity [8], the greening of suppliers [9], marker demand [10], customer and supplier collaboration [11], inter-firm R&D collaboration [12], ESG rating [13], the geographic proximity of financial resources [14], urban economy digitalization [15], pressure from stakeholders [16], and pressure from the government [17].

A small number of researchers investigated the influence of external sustainable identity on corporate green innovation. The reasons for examining institutional investors' (external) sustainable identity effects on green innovation are as follows. (1) Stakeholder theory suggests that institutional investors, as firms' vital stakeholders, play a crucial role in developing firms' green innovation. Researchers confirmed institutional investors' influencing effects on green innovation from various perspectives (For example, Ref. [18] confirms their influencing effects on green innovation from the perspective of their site visit, Ref. [19] from their ownership dispersion, and Ref. [20] from their portfolio characteristics). (2) However, in the growing importance of sustainable development, little research investigated the possible effect of their sustainable identity on corporate green innovation. Concerning institutional investor identity, organizational identity theory illustrates it as the core value and the beliefs that guide organizational behaviors [21]. Thus, if their investment goals and scopes (which influence their primary organizational behavior) include environmental and social governance, this article refers to it as a formed sustainable identity. This study determines such identity, based on the data of the institutional investor's investment goals and scopes as well as its shareholding ratio in companies. Based on this creatively constructed data and the above analyses, this study will answer the following research question: (1) Does the sustainable identity of institutional investors contribute to green innovation?

To answer this question, we examine the financial and fund data of Chinese-listed companies from 2010 to 2020. The financial and CSR reporting information is obtained from CSMAR and CNRDS databases. Bloomberg's ESG definition is used for determining institutional investors' sustainable identity. Considering that the sustainability of institutional investors is an exogenous factor, a difference-in-differences (DID) model is used to analyze the data.

Our result supports the assumption that institutional investors' sustainable identity can improve corporate green innovation. We use parallel trend, placebo, propensity score matching, difference-in-differences (PSM-DID), and changes in fixed effects to test the robustness of our conclusions. Additionally, we find that our benchmark result possesses heterogeneity in various corporate environmental responsibility performance components, where corporate environmental responsibility performance acts as a moderator between them. This means that the higher the corporate environmental performance, the stronger the influencing impact of institutional investors' sustainability on green innovation. Additionally, we observe that this external sustainable identity performs a shock effect similar to the sustainable rating from a third-party agency on corporate green innovation.

This study contributes significantly to the existing literature in two primary ways. First, it fulfills the research gap on green innovation's sustainability determinants by shifting the focus onto external factors from internal factors, commonly emphasized in prior works [19,22,23]. Second, it enriches the research on the relationship between institutional investors and green innovation by exploring their sustainable identity, while others [18,20,24,25] investigate their site visit, their shareholding, ownership dispersion, portfolio, and so on.

This work is closely related to [20,26,27] yet differs from them in the following ways. First, Ref. [20] focuses on an institutional investor's portfolio and investigates its influence on green innovation; meanwhile, this paper focuses on the green identity of institutional investors. Additionally, their work explores the overall social responsibility's heterogeneous effect on their relationship, while our work concentrates on specific environmental responsibility performance's heterogeneous and mechanism effect. Second, Refs. [26,27] examine institutional investor effects on corporate social responsibility or ESG performance, while our work concerns its effects on green innovation.

The following is this paper's remainder. The research hypotheses are proposed in Section 2. Section 3 presents the models, main variables, and data specifications. Section 4 reports the results of the benchmark regression analysis and robustness tests. Section 5 is the further analysis of heterogeneous, cross-sectional, and alternative factors. The final section concludes the paper.

## 2. Theoretical Background and Hypotheses Development

### 2.1. Green Innovation and Its Determinant FACTORS

Green innovation is one of the important ways to achieve sustainable development, but it leads to various benefits. Its economic benefits include saving resources [28], reducing production and process costs [29], green consumption behavior [28], and greater financial return and market share [29,30]. Its social benefits have good environmental performance [31], environmental legitimacy [32,33], and reputation building [34–36]. Additionally, due to its time consumption and short-term uncertainty, green innovation has the demerits of invisible or long-term return [24,29]), less financial benefit compared to its social benefits, and so on. Thus, investors' support possesses a considerable significance for green innovation development.

Prior research analyzed its determinants from both internal and external perspectives: its internal factors include green absorptive capacity [6], green organizational identity [7], board gender diversity [8], climate change exploration [37], resource and knowledge sharing with outside partners [38], and environmental policy [39]; and its external factors include the greening of suppliers [9], market demand [10], customer and supplier collaboration [11], inter-firm R&D collaboration [12], ESG rating [13], the geographic proximity of financial resources [14], urban economy digitalization [15], pressure from stakeholders [16], and institutional investors [20]. However, no research examines it from the perspective of external sustainability (which refers to institutional investor's sustainable identity in this paper).

### 2.2. Green Innovation and Institutional Investors

The reason for the search for sustainable identity effects from institutional investors is as follows. Institutional investors are important stakeholders with different characteristics who can influence corporate decision-making through active supervision and governance. Although institutional investors in China have a smaller influence as compared to those in developed markets, researchers are investigating its effect on companies' sustainable development [20,40,41]. For example, Ref. [18] investigates their relationship with green innovation from the effects of an institutional investor's site visit (field trips to a firm's headquarters and its operation facilities and their shareholding on green innovation). Moreover, [19] validates institutional investors' promoting effect on eco-innovation from their ownership dispersion (participation in a company's stock capital with different time horizons). Additionally, [20] states that institutional investors' portfolio characteristics (supervisory motivation and governance capability) are critical for corporate green innovation development.

Although numerous works examined green innovation determinant factors, including that of institutional investors from their various characteristics, there is limited knowledge regarding the influence investors' concern for sustainability has on it. The possible research gap is that it remains unclear whether their sustainable identity contributes to corporate green innovation.

### 2.3. Green Innovation and Institutional Investors' Sustainable Identity

First, stakeholder theory suggests that not only does major corporate decision-making, such as green innovation, need institutional investors' support [25,42,43], but institutional investors also need incentives to influence green innovation performance [18,24]. Institutional investors with sustainable involvement reached a consensus on issues such as long investment cycles in green innovation and, therefore, have a desire for long-term stable investment, which, in turn, belies a strong desire to supervise green innovation decision-making. This enables them to reduce the risks caused by information asymmetry and weakens the management's damage to green innovation development for short-term benefits. Institutional investors' sustainable involvement can become the driving force for corporate green innovation. Thus, it is important to investigate the sustainability of institutional investors for green innovation development.

Second, sustainable institutional investors will respond less sensitively to quantitively mispricing signals [44] and are more risk-tolerant. Moreover, investor preferences can change the original utility function, which can change companies' risk tolerance and investment budget [45]. Additionally, sustainable institutional investors can better supervise corporate sustainable development in fulfilling their responsibility toward shareholders and reap greater commercial performance [46]. These characteristics are conducive to the green innovation development of companies, which requires big initial capital investments with high uncertainty and whose profit cycle is relatively long [47]. Thus, we propose our hypothesis:

**H1.** *Institutional investors' sustainable identity promotes green innovation.*

## 3. Research Design

### 3.1. Model Setting

Based on the above arguments of sustainable institutional investors' behavior characteristics, our benchmark regression analysis examined the influencing effect of company institutional investors' sustainable identity on green innovation by using a two-way fixed effects staggered DID method, following prior research [48–51]. DID can be used to understand potential causal relationships [52], and institutional investors are exogenous variables, for which DID is a useful technique [48]. The sample constitutes Chinese-listed non-financial companies from 2010 to 2020. The regression model (1) is as follows:

$$GI = \alpha_0 + \alpha_1 IIssid_{it} + \delta Controls_{it} + Firm_i + Year_t + \varepsilon_{it} \tag{1}$$

*GI* represents corporate green innovation performance. $IIssid_{it}$ is the dependent variable, representing company institutional investors' sustainable identity. It equals 1 when companies have sustainable-identified institutional investors and 0 otherwise. $\alpha_1$ is the coefficient of institutional investors' sustainable identity and green innovation. $Controls_{it}$ is the control variable (*Size, Cash, Board, Holder, Supervisor, State, Indcd*) related to company green innovation. $Firm_i$ and $Year_t$ represent firm- and year-fixed effects.

### 3.2. Measures

Green innovation, the dependent variable, is the growth rate of green patent citations. Previous studies measured green innovation by companies' green product innovation and green process innovation. Most researchers use green patents as indicators of green innovation, as it is hard for Chinese companies to obtain eco-labeled product certifications and pursue green research and development.

Institutional investors' sustainable identity, the independent variable, is measured based on Ref. [48]'s work. The investor's target has a significant influence on their utility function and their risk and uncertainty tolerance, and thereby, the time tolerance of green innovation. Institutional investors are important shareholders of corporations. Therefore, it is important to examine the role of institutional investors' sustainable identity by their target and scope.

We constructed institutional investors' sustainable identity indicators as follows. First, we selected institutional investors from the list of the top ten shareholders of the company, who included ESG keywords in their investment scope and objectives. If their investment scope or objectives included any of the keywords associated with ESG rating standards (Bloomberg), they were defined as sustainable-identified institutional investors. Second, the impacts of their sustainability were determined by a dummy variable, which equaled 1 if the corporation had sustainable institutional investors; otherwise, it was 0.

Our control variables include $Size_{it}$, $Cash_{it}$, $State_{it}$, $Board_{it}$, $Holder_{it}$, $Supervisor_{it}$, and $Indcd_{it}$.

Larger firms' green activities invite investments from institutional investors and influence other companies' behavior [53]. Thus, we controlled for company size. Green

innovation's initial investment was large, and it needed a longer cycle to generate profits. Therefore, we controlled for net cash flow generated from operating activities/current assets (cash), as outperforming firms are more likely to engage in green innovation [54]. We controlled for company ownership nature (state) as state-owned and private-owned firms engage in sustainable development differently [55]. The factors of equity concentration (holder), board size (board), and supervisory board size (supervisor) are controlled, considering that a larger board of directors acquires more information and resources for green innovation [56] and the higher equity concentration means more power by the largest shareholders in terms of green innovation [57]. Table 1 lists the definitions of all variables.

**Table 1.** Variable definitions.

| Symbols | Definitions |
|---------|-------------|
| GI | Growth rate of green patent citations |
| Treat | A dummy variable: 1 if the company has a sustainable institutional investor during the sample period; 0 otherwise. |
| IIssid | A dummy variable: 1 if in the current year and after the company included a sustainable-identified institutional investor; 0 otherwise. |
| Size | Ln (1 + Total assets) |
| Cash | Net cash flow generated from operating activities/current assets |
| Holder | Largest shareholder's shareholding percentage |
| Board | Natural logarithm of board directors |
| Supervisor | Natural logarithm of supervisory board directors |
| State | A dummy variable: 1 for state-owned companies; 0 otherwise |
| ERP | Sum of WasteGasEmissRed, WasteWaterEmissRed, SootDustRed, SolidWasteDispUtil, NoiseLightRadGovern, and ClearProdImplement |

### 3.3. Data Collection

We used 2010 to 2020 data from a Chinese-listed company. The sample does not include data from the financial and real estate industries. Sustainable institutional investors' data, including their investment scope and target, were collected from institutional funds' annual reports and listed companies' top 10 shareholders' documents. This study used data from CSMAR and CNRDS. Table 2 displays descriptive statistics of the above variables.

**Table 2.** Descriptive statistics.

| Variable | Obs | Mean | Std. Dev. | Min | Max |
|----------|-----|------|-----------|-----|-----|
| Treat×IIssid | 24,344 | 0.2359103 | 0.4245751 | 0 | 1 |
| IIssid_value | 24,344 | 0.2046085 | 0.7359048 | 0 | 3.02567 |
| Size | 24,344 | 22.10897 | 1.341682 | 14.94164 | 28.63649 |
| Cash | 24,344 | 0.1169017 | 0.3529241 | −10.22522 | 35.17631 |
| Holder | 24,344 | 0.3365103 | 0.147225 | 0.0029 | 0.8999 |
| Board | 24,344 | 2.221691 | 0.3087126 | 0 | 3.367296 |
| Supervisor | 24,344 | 1.3919 | 0.3766214 | 0 | 3.367296 |
| State | 24,344 | 0.3560631 | 0.478844 | 0 | 1 |
| Indcd | 24,344 | 6.439821 | 3.971191 | 1 | 22 |
| ERP | 24,344 | 1.707074 | 2.499477 | 0 | 12 |

## 4. Analysis and Results

### 4.1. Regression Results

Table 3 shows the benchmark regression analysis results based on Equation (1). Column (1) reports the results with year- and firm-fixed effects, excluding the control variables. Column (2) reports the results with fixed effects, including the control variables. The regression coefficient ($\alpha_1$) is significantly positive ($\alpha_1 = 0.009$, t = −0.004, $p < 0.05$). This result shows that the company's institutional investors' sustainable identity is related to a 3.5% increase in the growth of green patent citation, indicating that companies' institu-

tional investors' sustainable identity enhances their green innovation. The R-squares in the regression analysis results are 0.388 and 0.389, respectively.

**Table 3.** Institutional investors' ESG involvement and green innovation.

| Variables | (1) DID GI | (2) DID GI | (3) OLS GI | (4) OLS GI |
|---|---|---|---|---|
| Treat×IIssid | 0.009 ** | 0.009 ** | | |
| | −0.004 | −0.004 | | |
| IIssid_value | | | 0.009 ** | 0.010 ** |
| | | | −0.005 | −0.005 |
| Size | | 0.015 *** | | 0.015 *** |
| | | −0.003 | | −0.003 |
| Cash | | −0.012 * | | −0.012 |
| | | −0.007 | | −0.007 |
| Holder | | −0.024 | | −0.024 |
| | | −0.026 | | −0.026 |
| Board | | −0.008 | | −0.008 |
| | | −0.007 | | −0.007 |
| Supervisor | | 0.01 | | 0.01 |
| | | −0.006 | | −0.006 |
| State | | −0.001 | | −0.001 |
| | | −0.011 | | −0.011 |
| Indcd | | −0.001 | | −0.001 |
| | | −0.001 | | −0.001 |
| Constant | 0.114 *** | −0.188 ** | 0.116 *** | −0.190 ** |
| | −0.002 | −0.075 | −0.001 | −0.075 |
| Observations | 23,841 | 23,688 | 23,841 | 23,688 |
| R-squared | 0.388 | 0.389 | 0.388 | 0.389 |
| Fixed effect | YES | YES | YES | YES |
| Controls | NO | YES | NO | YES |

Note: This table shows the link between institutional investors' sustainability and green innovation, with those of DID in Columns (1) and (2) and those of OLS regression analysis in Columns (3) and (4). All variables are defined in Table 1. * $p < 0.1$, ** $p < 0.05$, *** $p < 0.01$. Standard errors in parentheses.

*4.2. Robustness Tests*

Parallel trend, placebo, PSM-DID, change of, and dynamic fixed effects are used to test robustness. This section introduces their model setting and analysis results.

4.2.1. Parallel Trend Test

Based on a preliminary examination of the effect of institutional investors' sustainable identity in enhancing green innovation, if the above results can pass the parallel trend test, our assessment by the staggered DID technique can be considered reliable. We validate this by setting dummy variables and pursuing the year-by-year effects of institutional investors' sustainability on their green innovation. The model below is built to analyze such effects, following prior research [49].

$$GI = \alpha + \beta_1 Treat \times IIssid_{it}^{-4} + \beta_2 Treat \times IIssid_{it}^{-3} + \cdots + \beta_5 Treat \times IIssid_{it}^{current} + \cdots + \beta_{10} Treat \times IIssid_{it}^{5} + \delta Controls_{it} + Firm_{it} + Year_{it} + \varepsilon_{it} \qquad (2)$$

Treat $\times$ IIssid$_{it}^{relative\ year}$ is the dummy variable. Treat $\times$ IIssid$_{it}^{-P}$ is 1 for companies in the pth year after sustainable institutional investor involvement; otherwise, it is 0. We observe the coefficient for three years before and five years after sustainable institutional investors' involvement. We use the same control and fixed effects vectors as those in the benchmark regression analysis.

As Table 4 and Figure 1 indicate, the regression results are significantly positive at and after the present year. This indicates that companies' green innovation increases

immediately after sustainable institutional investors' involvement. Thus, the regression analysis results passed the parallel trend measurement.

**Table 4.** Parallel trend results.

| Variables | GI |
|---|---|
| $Treat \times IIssid_{t-3}$ | 0.014 |
| | −0.024 |
| $Treat \times IIssid_{t-2}$ | 0.029 |
| | −0.023 |
| $Treat \times IIss_{t-1}$ | 0.034 |
| | −0.022 |
| $Treat \times IIssid_{current}$ | 0.033 * |
| | −0.019 |
| $Treat \times IIssid_{t+1}$ | 0.039 * |
| | −0.02 |
| $Treat \times IIssid_{t+2}$ | 0.065 *** |
| | −0.021 |
| $Treat \times IIssid_{t+3}$ | 0.053 ** |
| | −0.021 |
| $Treat \times IIssid_{t+4}$ | 0.041 ** |
| | −0.021 |
| $Treat \times IIssid_{t+5}$ | 0.039 * |
| | −0.021 |
| Cash | −0.01 |
| | −0.007 |
| Holder | −0.02 |
| | −0.026 |
| Board | −0.007 |
| | −0.007 |
| Supervisor | 0.01 |
| | −0.006 |
| State | 0.002 |
| | −0.011 |
| Indcd | −0.001 |
| | −0.001 |
| Constant | 0.112 *** |
| | −0.02 |
| Observations | 23,688 |
| R-squared | 0.389 |

Note: This table shows parallel trend test results. All variables are defined in Table 1. * $p < 0.1$, ** $p < 0.05$, *** $p < 0.01$. Standard errors in parentheses.

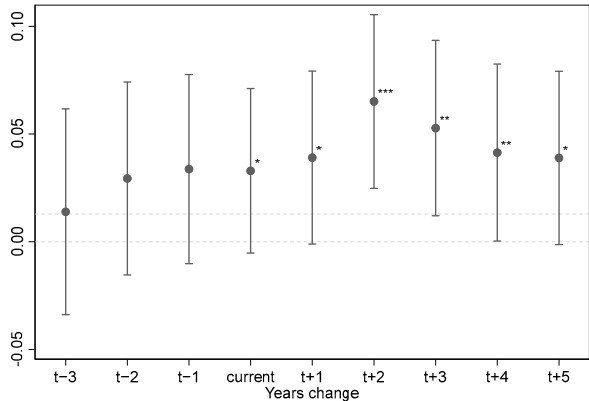

**Figure 1.** Parallel trend results. Note: This figure shows parallel trend test results. * $p < 0.1$, ** $p < 0.05$, *** $p < 0.01$.

### 4.2.2. Placebo Test

To validate the benchmark regression results produced by institutional investors' sustainable identity rather than other factors, a placebo test technique was applied to artificially alter the implementation time of sustainable institutional investors' involvement. We changed the timing of their involvement to four years in advance. If a placebo trial on an artificially changed variable cannot yield significant results, we can argue that the conclusion obtained from the previous benchmark regression results is correct. As shown in Table 5, the Treat×IIssid placebo was not significant, confirming the robustness of the benchmark regression.

**Table 5.** Placebo test.

| Variables | GI |
|---|---|
| Treat×IIssid_4 | 0.026 |
| | (0.019) |
| Size | 0.014 *** |
| | (0.003) |
| Cash | −0.012 * |
| | (0.007) |
| Holder | −0.023 |
| | (0.026) |
| Board | −0.007 |
| | (0.007) |
| Supervisor | 0.010 |
| | (0.006) |
| State | −0.001 |
| | (0.011) |
| Indcd | −0.001 |
| | (0.001) |
| Constant | −0.190 ** |
| | (0.075) |
| Observations | 23,688 |
| R-squared | 0.389 |
| Fixed effect | YES |
| Controls | YES |

Note: This table shows the placebo test results. All variables are defined in Table 1. * $p < 0.1$, ** $p < 0.05$, *** $p < 0.01$. Standard errors are in parentheses.

### 4.2.3. Selection Bias Test

Owing to the effects of selection bias of green innovation, firms with sustainable-identified institutional investors' involvement could exhibit higher values than those of firms without; therefore, comparing these two groups directly may not generate a fair result. To address sample selection bias, we use the PSM method to perform sample matching and DID to exclude bias. The sample matching process, analysis of the matching results, and regression analysis results are as follows.

First, the logit model is employed for the final propensity score, with the dummy variable for a company's institutional investors' sustainable identity (IIssid) as the dependent variable and company size (Size), Cash, State, Board, Supervisor, Holder, Industry (Indcd) as covariates. We select companies from the control group that possess characteristics similar to those in the treatment group, forming the resulting dataset for propensity score matching. Consequently, we obtained 18,129 untreated samples on support and 5640 treated samples on support using a 1:1 radius caliper (0.3066795) matching method.

Second, Table 6 shows that after matching, the standardized deviation of the independent variables drops from 89% to 22%, R2 from 0.040 to 0.001, and Medbias from 9.5 to 0.8, indicating that sample matching can overcome the issue of sample selectivity bias.

**Table 6.** PS test.

| Sample | Ps R2 | LR chi2 | MedBias | %Var |
|---|---|---|---|---|
| Unmatched | 0.040 | 1042.17 | 9.5 | 89 |
| Matched | 0.001 | 22.03 | 0.8 | 22 |

Third, Table 7 reviews the regression analysis outcome of the PSM-DID with fixed effects and control variables. The significantly positive coefficient $\alpha\_1$ ($p < 0.05$) indicates that companies' sustainable-identified institutional investors' involvement enhances green innovation.

**Table 7.** PSM-DID.

| Variables | GI |
|---|---|
| Treat×IIssid | 0.010 ** |
|  | (0.004) |
| Size | 0.015 *** |
|  | (0.003) |
| Cash | −0.017 * |
|  | (0.008) |
| Holder | −0.024 |
|  | (0.026) |
| Board | −0.007 |
|  | (0.007) |
| Supervisor | 0.010 |
|  | (0.006) |
| State | −0.000 |
|  | (0.011) |
| Indcd | −0.001 |
|  | (0.001) |
| Constant | −0.196 ** |
|  | (0.077) |
| Observations | 23,614 |
| R-squared | 0.389 |
| Fixed effect | YES |
| Controls | YES |

Note: This table shows the PSM-DID test results. All variables are defined in Table 1. * $p < 0.1$, ** $p < 0.05$, *** $p < 0.01$. Standard errors in parentheses.

### 4.2.4. Changes in Fixed Effects

Our benchmark regression controls for year- and firm-fixed effects. For robustness, this section reserves these two fixed effects and some other control factors, such as city–, industry–, province–, and year–industry interaction, as well as year–province interaction and year–city interaction fixed effects. The reason for controlling the fixed effects of cities and provinces is that the regional development level is one of the critical external environment conditions for business [2]. The regression analysis results are all significant ($p < 0.01$; Table 8), indicating that those fixed effects do not change the significance of the benchmark regression analysis results.

**Table 8.** Change of fixed effects.

| Variables | (1) GI | (2) GI | (3) GI | (4) GI | (5) GI |
|---|---|---|---|---|---|
| Treat×IIssid | 0.026 *** | 0.025 *** | 0.027 *** | 0.020 *** | 0.023 *** |
| | (0.004) | (0.004) | (0.004) | (0.004) | (0.004) |
| Size | 0.024 *** | 0.028 *** | 0.025 *** | 0.030 *** | 0.025 *** |
| | (0.002) | (0.001) | (0.002) | (0.002) | (0.002) |
| Cash | −0.046 *** | −0.023 *** | −0.051 *** | −0.020 *** | −0.050 *** |
| | (0.007) | (0.007) | (0.007) | (0.007) | (0.007) |
| Holder | −0.043 *** | −0.017 | −0.043 *** | −0.027 ** | −0.051 *** |
| | (0.013) | (0.012) | (0.013) | (0.012) | (0.013) |
| Board | −0.006 | −0.004 | −0.009 | −0.001 | −0.007 |
| | (0.006) | (0.006) | (0.006) | (0.006) | (0.007) |
| Supervisor | −0.011 ** | −0.005 | −0.009 * | 0.002 | −0.000 |
| | (0.005) | (0.005) | (0.005) | (0.005) | (0.006) |
| State | −0.025 *** | −0.000 | −0.024 *** | −0.003 | −0.027 *** |
| | (0.005) | (0.004) | (0.004) | (0.004) | (0.004) |
| Indcd | −0.003 *** | | −0.002 *** | | −0.002 *** |
| | (0.000) | | (0.000) | | (0.000) |
| Constant | −0.339 *** | −0.492 *** | −0.361 *** | −0.535 *** | −0.389 *** |
| | (0.035) | (0.032) | (0.034) | (0.033) | (0.035) |
| Observations | 21,197 | 23,850 | 21,201 | 23,843 | 21,201 |
| R-squared | 0.065 | 0.071 | 0.027 | 0.089 | 0.047 |
| City fixed effect | YES | | | | |
| Industry fixed effect | | YES | | | |
| Province fixed effect | | | YES | | |
| Year–industry fixed effects | | | | YES | |
| Year–province fixed effects | | | | | YES |
| Controls | YES | YES | YES | YES | YES |

Note: This table shows the results of the changed fixed effects. All variables are defined in Table 1. * $p < 0.1$, ** $p < 0.05$, *** $p < 0.01$. Standard errors in parentheses.

## 5. Further Analysis

### 5.1. Corporate Environmental Responsibility, Institutional Investor's Sustainable Identity, and Green Innovation

Stakeholders are paying more and more attention to corporate social and environmental responsibility, so companies with hypocritical environmental and social responsibility are at risk of losing their trust and undermining their legitimacy [58]. Thus, this paper further analyzes whether the performance of corporate environmental responsibility affects the influencing relationship between institutional investors' sustainable identity and green innovation.

Firstly, corporate environmental responsibility performance can reduce the problem of information asymmetry between institutional investors and enterprises. According to Ref. [59], when there is information asymmetry between investors and corporations, the disclosure of environmental responsibility performance information can help institutional investors evaluate the performance of enterprises in fulfilling their environmental responsibilities and then make decisions based on their investment nature. Corporations with high environmental responsibility performance attract more institutional investors with sustainable behavioral goals. Furthermore, these institutional investors have a positive impact on corporate sustainable development (such as green innovation). For example, companies add R&D involvement to meet environmental regulation requirements from institutional investors. Such activities are conducted using green patents and can result in green innovation. Therefore, we can argue that corporations with higher environmental responsibility performance attract more institutional investors with sustainable identities, which, in turn, supervise and promote the fulfillment of corporate environmental responsi-

bility and sustainable development. Thus, in the following two sections, we empirically test our assumption.

### 5.2. Heterogeneous Test

This section verifies the heterogeneous relationship between institutional investors' sustainable identity and green innovation. We expect the promotional effect of institutional investors' sustainable identity on green innovation to be significant in companies that better fulfill their environmental responsibility. We categorized our sample into two groups on the corporate environmental responsibility performance (ERP) and tested their regression results.

ERP can be measured by environmental information disclosure, which is an important manifestation of a company's environmental and social responsibility [60]. ERP is the sum of waste gas emission treatment, wastewater emission treatment, soot and dust treatment, solid waste treatment, noise light rad governance, and clear production implementation. Table 9 (1) displays the positively significant coefficient results of institutional investors' sustainable identity on green innovation at the 10% level ($\alpha$ = 0.013; t = $-0.007$) among firms within the high ERP group. Column (2) shows that the coefficient effect is not significant in the low group. This suggests that the ERP may serve as a moderator. Accordingly, the following section tests its moderating effect.

**Table 9.** Heterogeneous analysis.

| | (1) | (2) |
|---|---|---|
| | **High-ERP** | **Low-ERP** |
| **Variables** | **GI** | **GI** |
| Treat×IIssid | 0.013 * | 0.008 |
| | −0.007 | −0.005 |
| Size | 0.017 ** | 0.015 *** |
| | −0.008 | −0.004 |
| Cash | −0.042 ** | −0.008 |
| | −0.018 | −0.009 |
| Holder | −0.014 | −0.015 |
| | −0.051 | −0.033 |
| Board | 0.004 | −0.021 ** |
| | −0.013 | −0.009 |
| Supervisor | 0.009 | 0.014 * |
| | −0.011 | −0.008 |
| State | −0.02 | −0.007 |
| | −0.021 | −0.014 |
| Indcd | −0.002 | −0.001 |
| | −0.003 | −0.001 |
| Constant | −0.258 | −0.186 ** |
| | −0.187 | −0.09 |
| Observations | 8073 | 15,195 |
| R-squared | 0.43 | 0.43 |
| Fixed effect | YES | YES |
| Controls | YES | YES |

Note: This table shows the heterogeneous results among the high- and low- environmental responsibility performance groups. All variables are defined in Table 1. * $p < 0.1$, ** $p < 0.05$, *** $p < 0.01$. Standard errors in parentheses.

### 5.3. Cross-Sectional Analysis

After obtaining the benchmark result implying that institutional investors' sustainable identity enhances green innovation in Equation (1), we examined the moderating role of corporate environmental performance in this promotional relationship. Therefore, we added the corporate environmental performance variable and its interaction term with sustainable-identified institutional investors' input to the benchmark regression analysis, establishing Equation (3). This moderating variable is delegated by corporate environmen-

tal responsibility performance (ERP). Table 10 shows the relationship among institutional investors' sustainable identity, ERP, and green innovation.

$$GI = \beta_0 + \beta_1 Treat \times IIssid_{it} + \beta_2 ERP_{it} + \beta_3 Treat \times IIssid \times ERP_{it} + \delta Controls_{it} + Firm_i + Year_t + \varepsilon_{it} \qquad (3)$$

**Table 10.** Corporate environmental performance' moderating effects.

| | (1) | (2) |
|---|---|---|
| **Variables** | **GI** | **GI** |
| ERP | −0.001 | −0.001 |
| | (0.001) | (0.001) |
| Treat×IIssid×ERP | 0.003 ** | 0.003 *** |
| | (0.001) | (0.001) |
| Size | | 0.015 *** |
| | | (0.003) |
| Cash | | −0.012 * |
| | | (0.007) |
| Holder | | −0.024 |
| | | (0.026) |
| Board | | −0.008 |
| | | (0.007) |
| Supervisor | | 0.010 |
| | | (0.006) |
| State | | −0.001 |
| | | (0.011) |
| Indcd | | −0.001 |
| | | (0.001) |
| Constant | 0.117 *** | −0.188 ** |
| | (0.002) | (0.075) |
| Observations | 23,841 | 23,688 |
| R-squared | 0.388 | 0.389 |
| Fixed effect | YES | YES |
| Controls | NO | YES |

Note: This table shows the cross-sectional analysis results. All variables are defined in Table 1. * $p < 0.1$, ** $p < 0.05$, *** $p < 0.01$. Standard errors in parentheses.

Columns (1) and (2) in Table 10 report the results moderated by ERP: the former excludes the control variables while the latter includes them. The interacting coefficient $\beta_3$ is significantly positive ($p < 0.05$). This indicates that the greater the ERP, the stronger the influence of institutional investors' sustainable identity on green innovation performance.

*5.4. Alternative Factor Test*

Given that the chosen sustainable identity in this paper is an external factor, we would like to confirm whether it has a mutually exclusive effect with other external sustainability factors. As external sustainability acts as a shock effect, we choose a sustainable rating from agencies as the alternative. We re-examined our benchmark regression under two conditions: the sample group with ESG rated by an agency and that without ESG rated by an agency. As Table 11 shows, the benchmark regression result did not change in the group without the rating shock, while it did change in the group with the rating shock. This result suggests the effect of two external sustainability as an alternative for corporate sustainable development.

**Table 11.** Regression results under alternative conditions.

| Variables | (1) Without Rating Shock GI | (2) With Rating Shock GI |
|---|---|---|
| Treat×IIssid | 0.012 ** | 0.005 |
| | (0.005) | (0.006) |
| Size | 0.016 *** | 0.006 |
| | (0.004) | (0.006) |
| Cash | −0.005 | −0.020 * |
| | (0.010) | (0.011) |
| Holder | −0.012 | −0.040 |
| | (0.034) | (0.044) |
| Board | −0.017 * | 0.005 |
| | (0.009) | (0.012) |
| Supervisor | 0.017 ** | −0.002 |
| | (0.008) | (0.011) |
| State | 0.004 | −0.024 |
| | (0.014) | (0.018) |
| Indcd | −0.001 | −0.000 |
| | (0.001) | (0.002) |
| Constant | −0.228 ** | 0.026 |
| | (0.095) | (0.150) |
| Observations | 14,594 | 9051 |
| R-squared | 0.396 | 0.410 |
| Fixed effect | YES | YES |
| Controls | YES | YES |

Note: This table shows the regression results under different conditions. All variables are defined in Table 1. * $p < 0.1$, ** $p < 0.05$, *** $p < 0.01$. Standard errors in parentheses.

## 6. Discussion and Conclusions

### 6.1. Discussion

Our study seeks to explore whether institutional investor's sustainable identity influences corporate green innovation. Based on stakeholder theory and organizational identity theory and applying the difference-in-differences (DID) design, we validate that institutional investors' sustainable identity has a promoting effect on corporate green innovation. This promoting effect is more pronounced in companies that perform better in environmental responsibility and act as a moderator between them. In further analysis, we find the effects of this external sustainable identity and the sustainable rating of the third-party agency to be mutual alternatives for enhancing corporate green innovation.

Despite being consistent with the results of Refs. [20,26,27] on the relationship between institutional investors and green innovation, our study differs from these studies in the following ways: (1) Ref. [20] focuses on the influencing effect of institutional investors' portfolios on green innovation, while this paper is based on institutional investor's green identity, drawing upon the insights of Ref. [61]. Additionally, our work differs from their works on mechanism effects investigation: their work focuses on the heterogeneous effect of overall social responsibility, while our work is on the specific environmental responsibility performance's heterogeneous and mechanism effect. (2) Both Refs. [26,27] examine institutional investor's effects on corporate social responsibility or ESG performance, while our work is interested in its effects on green innovation.

### 6.2. Theoretical Contributions

This study contributes significantly to the existing literature in two primary ways. First, our findings fill the research gap of searching for green innovation's determinates from an external sustainability perspective; this stream of research overlooks investigating the possible effect on corporate green innovation from sustainability efforts of institutional investors, mostly focusing on that from organizational green identity, the greening of suppliers,

financial resources, economic digitalization, and so on in prior works [19,22,23,62–64]. Second, it enriches the research on the relationship between institutional investors and green innovation by exploring their sustainable identity, while others [18,20,24,25] investigate their site visit, their shareholding, ownership dispersion, portfolio, and so on.

### 6.3. Practical Implications and Policy Suggestions

Our findings hold practical implications for various stakeholder groups. First, our research findings reveal the important role of investor's sustainable identity, such as ESG-targeted institutional investors, in enhancing corporate green innovation development. (1) This finding recommends firm management develop clear sustainable development strategies, strengthen communication with institutional investors, and showcase their efforts and achievements in sustainable development to enhance institutional investors' confidence in the sustainable development of the enterprise. (2) The result on the moderating effects of company environmental responsibility performance provides administrators valuable suggestions on their specific information disclosure guidelines. Managers, while pursuing economic benefits, must fully recognize the importance of environmental and social responsibility achievements, which eventually attract more institutional investors and further promote the company's green development. (3) It is essential for governments and regulators to establish robust frameworks for corporate environmental, social, and governance information disclosure, emphasizing the importance of green innovation. By facilitating collaboration between stakeholders and businesses, the authorities can effectively guide the trajectory of sustainable development.

### 6.4. Limitations and Future Research Directions

This study has some limitations. First, our sample was selected from Chinese-listed companies. Further studies are required to confirm our results using samples from other less-developed countries. Second, future work should examine the dynamic ESG results and green innovation using other databases, such as Syntao Finance. Third, additional perspectives are required to study the effects of other stakeholder's (e.g., retail investors) sustainability on ESG and green innovation in future works. Fourth, further study is required to test our results with data from 2021 to 2023. Therefore, this study is not the last to investigate exogenous sustainability.

**Author Contributions:** Conceptualization, X.Y. and C.Y.; Methodology, X.Y. and C.Y.; Validation, C.Y.; Formal analysis, X.Y. and C.Y.; Investigation, X.Y. and C.Y.; Resources, X.Y. and C.Y.; Writing—original draft, X.Y. and C.Y.; Writing—review & editing, X.Y. and C.Y.; Visualization, X.Y.; Supervision, X.Y.; Project administration, X.Y.; Funding acquisition, X.Y. All authors have read and agreed to the published version of the manuscript.

**Funding:** This research was supported by the Natural Science Foundation of Inner Mongolia (No. 2023LHMS07016) and the Fundamental Research Fund for the directly affiliated university of Inner Mongolia (No. 2022 JBQN056).

**Data Availability Statement:** No new data were created or analyzed in this study. Data sharing is not applicable to this article.

**Conflicts of Interest:** The authors declare no conflicts of interest.

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
