# Peer review of "The Impacts of External Sustainability: Institutional Investors’ Sustainable Identity, Corporate Environmental Responsibility, and Green Innovation"

_sustainability, doi:10.3390/su16051961_

Round 1

Reviewer 1 Report

Comments and Suggestions for Authors

I am delighted to evaluate this study that examines the research gap regarding the external sustainable consequences on business sustainable growth. The study examines the influence of institutional investors' commitment to sustainability and corporate environmental responsibility on green innovation. This is done by employing the difference-in-differences (DID) design and analysing data from Chinese listed businesses between 2010 and 2020.

This study makes a substantial contribution to the current body of research by addressing the gap in knowledge on the elements that determine the sustainability of green innovation. Unlike previous studies that focused on internal issues, this study examines external factors.  Furthermore, this study enhances the existing research on the correlation between institutional investors and environmentally friendly innovation by examining their sustainable identity, shareholding, ownership dispersion, portfolio composition, and other related factors.

The following theory is proposed: institutional investors' sustainable identity fosters the advancement of environmentally friendly innovation.

The paper has a smooth flow and effectively situates the reader within the context of the subject matter. The purpose and objective are clearly articulated and stem from the deficiencies in the topic matter. The paper's argument is constructed upon a suitable foundation of theory and concepts, and the methodology is well elucidated and facilitates replication.

The research findings demonstrate the significant impact of sustainable identities among investors, such as ESG-focused institutional investors, in promoting the development of environmentally friendly innovations within corporations. This discovery implies that the management of the company should formulate explicit strategies for sustainable development. To promote institutional investors' trust in the sustainable growth of the company, it is important to improve communication with them and highlight the enterprise's efforts and successes in sustainable development. Furthermore, the findings about the moderating impacts of a company's environmental responsibility performance offer administrators helpful recommendations for their precise information disclosure requirements. Managers must acknowledge the significance of environmental and social responsibility accomplishments in addition to pursuing commercial gains. These achievements will ultimately attract more institutional investors and further advance the company's green development.

The results and conclusions align with the remainder of the study and are thoroughly examined, emphasising certain practical aspects. It provides a clear identification of the consequences for research, practice, and society, effectively connecting the divide between theory and practice.

The work exhibits a comprehensive grasp of the pertinent literature on the topic and references a suitable array of literature sources pertaining to the study. The text effectively articulates its argument, in accordance with the specialised terminology of the discipline and the anticipated level of understanding of the journal's audience.

I feel this work can be published after undergoing a thorough proofreading of its English grammar.

Comments on the Quality of English Language

Minor Proofread of English grammar required

Author Response

Thanks for your kind comments. The reply to your comments is attached. 

Reviewer 2 Report

Comments and Suggestions for Authors

Please see the attached referee report.

Comments on the Quality of English Language

Please see the attached referee report.

Author Response

Thanks for your suggestions. The reply to your comments is attached. 

Reviewer 3 Report

Comments and Suggestions for Authors

I have a few recommendations:

In the theoretical part, the authors should add more references and engage with them, presenting at least some of them in more detail. 

The references should use the guidelines of the journal for both citations in the text and at the end. 

The method and the results are well presented and structured. The 5th section should be Discussions where you should present your results in comparison to other studies in the literature. 

The Conclusions should be section number 6, not 8. After limitations you should add a phrase to mention some future research directions that you or other researchers can use, to expand the research or to correct the limitations. 

You have only one hypothesis, which is very limited so you might include other future research directions that might consider other possible hypotheses. 

Author Response

Thanks for your suggestions. Reply for your comments is attached. 

Reviewer 4 Report

Comments and Suggestions for Authors

Abstract

 1.   In the abstract part of the manuscript, 1-2 sentences are used to describe the research background and the limitations of previous studies.

 2.  The enlightenment and significance of the summary are not clear and specific.

Introduction

3.   The introduction lacks a clear theoretical basis.

4.   This study will answer two research questions: (1) Does institutional investors’ sustainable identity contribute to green innovation? The introduction only raises one question, may I ask where is the other research question?

5.   Since the acronym "CSR" in this part appears for the first time in the manuscript, its full name should be stated and explained accordingly.

Literature review

 6.   The literature review did not point out the limitations of scholars' research in this field, lacking clear academic logic and criticism.

Methodology and data

 7.   Lack of explanation of the merits, applicability and reasons for not choosing alternative methods.

8. Some important literature using the difference-in-differences approach has been omitted.
e.g., Abudu, H., Wesseh Jr, P. K., & Lin, B. (2023). Are African countries on track to achieve their NDCs pledges? Evidence from difference-in-differences technique. Environmental Impact Assessment Review, 98, 106917.

Li, X., Huang, Y., Li, X., & Liu, X. (2023). Mechanism of smart city policy on the carbon emissions of construction enterprises in the Yangtze River Economic Belt: a perspective of the PESTEL model and the pollution halo hypothesis. Humanities and Social Sciences Communications, 10(1), 580.

9.   No reference is given in the control variables section.

10.   You are advised to use pictures to check parallel trends, which are clearer and more intuitive.

Discussion

 11.   It is suggested that the author add a discussion section after the result analysis: rediscuss whether the results of other similar studies are consistent with the results of the manuscript study, and analyze the reasons if they are inconsistent.

Conclusions

12.    In the conclusion part, 1-3 sentences should be used to describe the purpose, content, methods and summary of the manuscript. Please ask the author to supplement this part.

13.    The research conclusions are suggested to be divided into several points.

 14.   Please give targeted policy suggestions based on the conclusions of the manuscript study. In summary, the authors are advised to carefully revise this manuscript in light of the above comments. I sincerely look forward to receiving the revised version.

Author Response

Thanks for your suggestions. Reply to your comments is attached. 

Reviewer 5 Report

Comments and Suggestions for Authors

Title: Internal and external sustainability and corporate environmental responsibility

This paper is very interesting to read. I hope that my comments are useful to enhance the quality of the paper.

1. Abstract

Please be more specific about the research purpose and justify the theoretical gap. The abstract should be written in the order of research motivation, research gap, theoretical positioning (i.e., argument), methodology, and summary of results.

2. Introduction

The rationale behind the background of proposing the your framework remains unclear to me.  Please be more specific about the research purpose and justify the theoretical gap. It seems that there is a lack of theoretical explanation. It is because the gap identification is not explained clearly and is limited in the current form. Also, there is very little evidence to support the authors’ arguments on the gap justification. Could you please clarify the theoretical gaps that have not been discovered in the existing theories? What theory was employed in this study to fill the theoretical gap? Please try to tackle the existing theories and literature from which the one RQs suggested by the authors were derived, and present the evidence. It is necessary to rewrite the introduction by distinguishing what we know from what we do not know and theorizing it.

3. Theoretical development and hypotheses

There are difficulties with reading and following the logic due to the lack of theoretical lens. The literature review should be conducted to the extent that hypotheses can be presented in a nuanced manner, but this manuscript is somewhat lacking in that respect. Your hypothesis's logical sentencing is not a decent; it should be emerged from the theory or theoretical void. Please refer to the following literature to strengthen the hypothesis design.

For sustainable or green innovation

 - Unpacking the sustainable performance in the business ecosystem: Coopetition strategy, open innovation, and digitalization capability, Journal of Cleaner Production, 412, 137433. 

 - Digitalization capability and sustainable performance in emerging markets: Mediating roles of in/out-bound open innovation and coopetition strategy, Management Decision.

4. Conclusion

Discussions are also weakly developed. there are actually no discussions only implications are there. You need a separate section or subsection for that. It needs way more work. Please mention how the work is different from previous works and which findings are in line with the literature and what is their significance. Try to discuss this once you will mention research questions. So map discussions with RQs and hypotheses.

In addition, the identification of the theoretical contribution within this study presents challenges. It is imperative to elucidate the specific enhancements that have been introduced to the extant literature. Nonetheless, it is noteworthy that the existing theoretical implications demonstrate a degree of sufficiency in facilitating meaningful connections with prior research endeavors.

I anticipate incorporating the cited papers enumerated below within the concluding segment, subsequent to a thorough contemplation regarding their potential contribution to enriching the research context. The decision to assimilate or employ these references within your study is at your discretion; you retain the prerogative to disregard them as well, The effects of open innovation on eco-innovation in meta-organizations Evidence from Korean SMEs, Asian Business & Management, Paving a way toward green world: Two-track institutional approaches and corporate green innovation, IEEE Transactions on Engineering Management.

Comments on the Quality of English Language

Please use english language editing.

Author Response

(The authors gave the same response as above.)

Round 2

Reviewer 2 Report

Comments and Suggestions for Authors

On the basis of the authors’ substantial revision and improvement, I would like to recommend acceptance after minor revisions for the writing. I will report the writing imperfection in specific-comment part. Please see my referee report which is chosen by the review system.

Comments on the Quality of English Language

On the basis of the authors’ substantial revision and improvement, I would like to recommend acceptance after minor revisions for the writing. I will report the writing imperfection in specific-comment part. Please see my referee report which is chosen by the review system.

Author Response

Thanks for your comments. I have revised the manuscript as you suggested, and Reply file is attached. 

Reviewer 4 Report

Comments and Suggestions for Authors

The authors have carefully revised the manuscript, and the present version is acceptable.

Author Response

Thank you so much for your help.

Reviewer 5 Report

Comments and Suggestions for Authors

I would like to extend my sincere thanks for the considerable effort and commitment the authors have devoted to this manuscript. The current version has markedly advanced in every aspect compared to its initial submission. The manuscript is now well positioned for publication.

Comments on the Quality of English Language

Please check the overall grammar and typos.

Author Response

Thank you so much for your help.  I have re-checked the English language and upload the revised version.